# Hippo Pathway in Mammalian Adaptive Immune System

**DOI:** 10.3390/cells8050398

**Published:** 2019-04-30

**Authors:** Takayoshi Yamauchi, Toshiro Moroishi

**Affiliations:** 1Department of Molecular Enzymology, Faculty of Life Sciences, Kumamoto University, Kumamoto 860-8556, Japan; yamauchi0@kumamoto-u.ac.jp; 2Center for Metabolic Regulation of Healthy Aging, Faculty of Life Sciences, Kumamoto University, Kumamoto 860-8556, Japan; 3Precursory Research for Embryonic Science and Technology (PRESTO), Japan Science and Technology Agency (JST), Kawaguchi 332-0012, Japan

**Keywords:** Hippo pathway, innate immunity, adaptive immunity, cancer immunity, autoimmunity, YAP (yes-associated protein), TAZ (transcriptional co-activator with PDZ-binding motif), LATS (large tumor suppressor kinase), MST (mammalian STE20-like protein kinase)

## Abstract

The Hippo pathway was originally identified as an evolutionarily-conserved signaling mechanism that contributes to the control of organ size. It was then rapidly expanded as a key pathway in the regulation of tissue development, regeneration, and cancer pathogenesis. The increasing amount of evidence in recent years has also connected this pathway to the regulation of innate and adaptive immune responses. Notably, the Hippo pathway has been revealed to play a pivotal role in adaptive immune cell lineages, as represented by the patients with T- and B-cell lymphopenia exhibiting defective expressions of the pathway component. The complex regulatory mechanisms of and by the Hippo pathway have also been evident as alternative signal transductions are employed in some immune cell types. In this review article, we summarize the current understanding of the emerging roles of the Hippo pathway in adaptive immune cell development and differentiation. We also highlight the recent findings concerning the dual functions of the Hippo pathway in autoimmunity and anti-cancer immune responses and discuss the key open questions in the interplay between the Hippo pathway and the mammalian immune system.

## 1. Introduction

Since its initial discovery in *Drosophila*, the Hippo pathway has gained immense attention for being strongly involved in organ development [1,2,3,4], stem cell biology [5,6,7], regeneration [8,9,10], and cancer biology [11,12,13,14]. The Hippo pathway responds to a wide range of extracellular and intracellular physiological cues, sensing the entire cellular environment, orchestrating cellular responses, and thus, contributing to cell fate determination [15,16]. The Hippo pathway is now known to be composed of more than 30 components, including the core kinase module and the transcriptional module [17]. As shown in Figure 1, the kinase module includes 11 kinases, namely mammalian STE20-like protein kinase 1 (MST1, also known as STK4) and MST2 (also known as STK3), mitogen-activated protein kinase kinase kinase kinases (MAP4Ks, including seven kinases, namely, MAP4K1/2/3/4/5/6/7), large tumor suppressor kinase 1 (LATS1) and LATS2, and in addition, their activating adaptor proteins, salvador family WW domain-containing protein 1 (SAV1), Ras-related proteins RAP2s (including three GTPases RAP2A/B/C), and MOB kinase activator 1A (MOB1A) and MOB1B, respectively. The transcriptional module includes yes-associated protein (YAP) and transcriptional co-activator with PDZ-binding motif (TAZ, also known as WWTR1), in combination with their best-characterized target transcription factors, TEA domain family members (TEADs, including four transcription factors, namely, TEAD1/2/3/4). When the kinase module is “ON” (the Hippo pathway is “activated”), MST1/2 or MAP4Ks phosphorylate and activate the downstream kinases LATS1/2, which in turn promotes inhibitory phosphorylation of the transcriptional co-activators YAP/TAZ, resulting in the cytoplasmic sequestration or proteasomal degradation of YAP/TAZ [18]. In contrast, when the upstream kinase module is “OFF” (the Hippo pathway is “inactivated”), hypophosphorylated YAP/TAZ translocate into the nucleus wherein they bind to and thus activate TEADs transcription factors to promote the target gene transcription. YAP/TAZ-mediated transcription generally drives multiple aspects of cell behavior, including cell proliferation, survival, cell plasticity, and stemness, which is essential for tissue development and regeneration [19]. Thus, in short, the activation of LATS1/2 kinases and the inactivation of YAP/TAZ transcriptional co-activators represent the major molecular functions of the canonical Hippo pathway.

In addition to its roles in tissue development and tumorigenesis, numerous studies in recent years have revealed the extensive roles of the Hippo pathway in immune regulation, both in adaptive and innate immune systems (Figure 2). For example, *Drosophila* Hippo (Hpo) and its mammalian homologues MST1/2 have been revealed to mediate Toll-like receptor (TLR) signaling in both flies [20] and mammals [21,22]. YAP/TAZ bind and inhibit TBK1 (TANK binding kinase 1) or IRF3 (interferon regulatory factor 3) to antagonize the antiviral innate immune responses [23,24]. The critical functions of the Hippo pathway in innate immune responses have also been reviewed elsewhere [25,26,27]. In this review, we focus on the current knowledge about the functions of the core Hippo pathway components in adaptive immunity, particularly in lymphocyte homeostasis during their development and differentiation. Although the molecular functions of MST1/2 in the mammalian adaptive immune system have been extensively studied in previous works, characterization of the other components of the Hippo pathway is an emerging field of research. In contrast to their pivotal functions in adherent cell physiology, it appears that YAP/TAZ are dispensable for physiological and malignant hematopoiesis [28]. Recent studies have demonstrated that MST1/2 regulate the lymphocyte biology independently of the key Hippo pathway components YAP/TAZ and LATS1/2 [29,30]. Indeed, growing evidence suggests a crosstalk between the Hippo pathway and other pivotal signaling networks involved in immune regulation, such as MAPK (mitogen-activated protein kinase), p53, and the FOXO (forkhead box O) pathway [17,31,32]. We also discuss the complexity of the signal transduction mechanisms downstream of the Hippo pathway in immune cells, which appear to be distinct from those in adherent cells that have been widely used to draw the Hippo signaling network to date. Although the Hippo pathway takes its name from MST1/2—the mammalian homologs of the *Drosophila* Hippo (Hpo), MST1/2 are also known to regulate several proteins other than the key Hippo signaling components. Therefore, the functional outputs of MST1/2 are not limited to LATS1/2 or YAP/TAZ [18]. In this review, we define the signaling that specifically regulates LATS1/2 kinase activity and/or YAP/TAZ transcriptional activity as the “canonical” Hippo pathway. Other signaling cascades that involve the core Hippo pathway components (particularly MST1/2) but do not regulate LATS1/2 kinase or YAP/TAZ are defined as the “alternative” Hippo pathway (Figure 1).

## 2. Hippo Pathway in Adaptive Immune Cell Lineage and Functions

Adaptive immunity is defined by antigen-specific immune responses, consisting of cellular (cell-mediated) and humoral (antibody-mediated) responses. All T cells, B cells, and antigen-presenting cells cooperatively orchestrate this process. Recent studies have revealed pivotal functions of MST1/2 in T-cell development and differentiation, as well as in B cell homeostasis in the splenic marginal zone and the periphery. MST1/2 also play key roles in antigen-presenting cells, including macrophages and dendritic cells. It is likely that MST1/2 regulate the adhesion and trafficking of the immune cells and thus coordinate various immunological events. Mechanistically, MST1/2 appear to exert their biological functions via both canonical and alternative downstream effectors in the Hippo pathway, as suggested by a series of in vivo mice studies.

### 2.1. Clinical Significance of the Hippo Pathway in Adaptive Immunity

The importance of the Hippo signaling pathway in the adaptive immune system is supported by the clinical case reports of patients who have loss of heterozygosity mutations of the MST1 gene or hyper-methylation of the MST1 promoter region [33,34,35,36,37,38]. MST1-deficient patients show susceptibility to bacterial and viral infections, clinical signs of T and B cell lymphopenia, and in addition, although counterintuitive, autoimmune manifestations (this seemingly contradictory outcome of MST1 deficiency is discussed in Section 2.4.). MST1/2-deficient mice were found to recapitulate some of these symptoms in the patients, including lymphopenia and autoimmune symptoms such as autoantibody production [29,33,36,39]. These mouse models have greatly helped in our understanding of the biological functions and molecular mechanisms of immune regulation by MST1/2 kinases.

### 2.2. MST1/2 in T Cell Development

Hematopoietic stem cells (HSC) give rise to both common lymphoid progenitor cells (CLP) and common myeloid progenitor cells (CMP) inside the bone marrow. CLP produce T cell progenitors, known as early thymic progenitors (ETP), which develop and mature within the thymus (Figure 3).

T lymphocytes originate in the thymus via three-steps, initiating as double-negative thymocytes (DN; CD4^−^CD8^−^), then changing to double-positive thymocytes (DP; CD4^+^CD8^+^), and finally, maturing into single-positive (SP; CD4^+^CD8^−^ or CD4^−^CD8^+^) thymocytes. Single-positive mature thymocytes are then released from the thymus to the peripheral tissues to compose and start adequate adaptive immune responses [40,41]. MST1/2 are found to express at low levels in double-positive thymocytes, but their expression rises at the single-positive stage [30], suggesting the important functions of MST1/2 in later stage of T cell development.

Using MST1 single knockout mice (*Mst1*^−/−^ mice), earlier studies have found the accumulation of *Mst1*^−/−^ single-positive thymocytes in the perivascular space where T cells exit into the periphery, resulting in a decreased number of peripheral T cells [42,43]. In vitro lymphocyte adhesion cascade assays demonstrated that *Mst1*^−/−^ T cells show reduced stopping times and disrupted integrin-mediated adhesion to endothelium, indicating that MST1 plays a major role in the efficient emigration of T cells from the thymus [42]. Mechanistically, the involvement of the alternative Hippo pathway has been proposed. MST1 in complex with RAP1 and RAPL (also known as RASSF5) phosphorylates NDR1 (nuclear Dbf2-related 1, also known as STK38) kinase, which recruits KINDLIN-3 (also known as FERMT3) to the immune synapse [44]. KINDLIN-3 was independently shown to mediate the high-affinity interaction between LFA-1 (also known as ITGB2; expressed on T cells) and ICAM1 (intercellular adhesion molecule 1; expressed on endothelial cells) to ensure efficient lymphocyte migration [45,46,47]. Another proposed mechanism suggests that MST1/2 directly phosphorylate and thus mediate MOB1A/B binding to DOCK8 (dedicator of cytokinesis 8) [30]. DOCK8 modulates the cytoskeletal remodeling as well as the cell migration in thymocytes by acting as a guanine nucleotide exchange factor (GEF) for RAC1 (Rac family small GTPase 1) [48]. It is to be noted that T cells from DOCK8-deficient mice shared some phenotypes of the MST1/2-deficient thymocytes, peculiarly a failure to polarize LFA-1 to the immune synapse [48], suggesting a key function of the Hippo pathway in small GTPase activation and cytoskeletal remodeling. Thus, collectively, MST1 functions in both cytoskeletal remodeling (through MOB1A/B phosphorylation) as well as cell adhesion (through NDR1 phosphorylation), ensuring the efficient emigration of T cells from the thymus.

### 2.3. Hippo Signaling in Effector T Cell Differentiation and Functions

Mature T lymphocytes that successfully pass the thymic selection incessantly migrate from the thymus towards the secondary peripheral lymphoid system to prepare for the antigen stimulation followed by the activation and differentiation phase (Figure 4). Naive T cells (Th0 cells) comprise CD4^+^ and CD8^+^ T cells that have not encountered its cognate antigen within the periphery and thus have not been differentiated. The expression levels of MST1 decrease as cell differentiation progresses from naive CD4^+^ T cells to effector/memory T cells. In contrast, MST2 expression levels remain constant [29], indicative of a dominant role of MST1 in naive T cell functions. Although the precise mechanism remains to be elucidated, recent studies have suggested that MST1, via the alternative Hippo pathway, contributes to the maintenance of naive CD4^+^ and CD8^+^ T cells. MST1 promotes cell survival [34,49] and restricts antigen receptor-induced proliferation [29] of naive T cells in order to maintain their homeostatic state. In addition to its role in thymic egress, MST1 functions in naive T cell survival and maintenance, may also account for T cell lymphopenia observed in human patients with MST1 deficiency.

Upon stimulation by the T cell receptor signaling and specific cytokines in the surrounding environments, naive CD8^+^ T cells become cytotoxic T lymphocytes (CTL), while naive CD4^+^ T cells activate and differentiate into specialized subtypes, including type 1 helper (Th1) cells, Th2 cells, Th17 cells, or regulatory T cells (Treg) (Figure 4). We highlight the emerging roles of the Hippo pathway in cytotoxic T cells as well as Th17 and Treg cells below.

#### 2.3.1. Hippo Pathway in Cytotoxic CD8^+^ T Cell Differentiation and Functions

Although much is still unknown about the roles of the Hippo pathway in the regulation of cytotoxic T cell physiology, it is known that the T-cell-receptor and IL-2 (interleukin 2) signaling induce the expression and activation of the canonical Hippo pathway in CD8^+^ T cells [50]. Activation of the T cell intrinsic Hippo signaling pathway suppresses YAP-mediated induction of the expression of transcriptional repressor BLIMP-1 (also known as PRDM1), thus resulting in terminal differentiation of the CD8^+^ T cell [50]. However, another report has suggested that MST1 in CD8^+^ T cells suppresses the cytotoxic function of CD8^+^ T cells upon T cell receptor stimulation [51]. MST1 via FOXO suppresses the expression of the transcription factor T-bet (also known as TBX21). T-bet boosts cytotoxic T cell functions by inducing the effector cytokine IFNγ (interferon gamma) and the cytotoxic protease granzyme B expression [52]. Thus, MST1 deletion in CD8^+^ T cells potentiates cytotoxic effector functions and prevents tumor growth in a syngeneic mouse model [51]. Together, these studies suggest seemingly paradoxical results that the Hippo pathway promotes terminal differentiation of CD8^+^ T cell while suppressing their cytotoxic functions. Future studies validating the role of other Hippo pathway components in cytotoxic T cells will be required in order to clarify the complex regulatory mechanisms of cytotoxic T cell differentiation and functions by the Hippo pathway.

#### 2.3.2. Hippo Pathway in Helper CD4^+^ Th17 and Treg Cells

Among CD4^+^ cell subsets, critical functions of the Hippo pathway in the regulation of Th17 and Treg cells have recently been revealed. IL-17-producing Th17 cells represent a pro-inflammatory subset, while FOXP3 (forkhead box P3)-expressing Treg cells are immunosuppressive [53,54]. The balance between Th17 and Treg cells is regulated by the inflammatory cytokines such as IL-6 and TGFβ (transforming growth factor beta), which has emerged as a striking factor in autoimmunity and cancer immune escape [53,54] (Figure 4).

Previous studies have demonstrated that MST1 promotes Treg differentiation and prevents autoimmunity and tissue damage [39,55]. MST1 appears to enhance the expression of FOXP3, the key transcriptional factor for Treg lineage, via the alternative Hippo pathway involving FOXO1/3 [39] or SIRT1 (sirtuin 1) [55]. In addition, another report suggested that MST1 also contributes to the contact-dependent suppression of Treg cells by mediating the immunological synapse formation between dendritic and Treg cells [56]. Collectively, these findings highlight the diverse functions of MST1 in Treg physiology.

A complementary recent study demonstrated that TAZ promotes Th17 lineage and inhibits Treg differentiation of naive CD4^+^ T cells [57]. On one hand, TAZ binds to RORγt (RAR related orphan receptor gamma t) and enhances its transcriptional activity to promote Th17 differentiation. On the other hand, TAZ competes with FOXP3 for its binding to the histone acetyltransferase TIP60 (also known as KAT5), thus destabilizing FOXP3 and inhibiting Treg differentiation [57]. Interestingly, the expression of TAZ is increased during Th17 differentiation in vitro [57]. This is in contrast with YAP, which is not expressed in Th17 cells, but is highly expressed in Treg cells [58]. Another study found that YAP in Treg cells induces the genes involved in the TGFβ superfamily member activin pathway and thus reinforces TGFβ–SMAD signaling, upregulating FOXP3 expression and Treg functions. Genetic inhibition of YAP dramatically dampened the immune suppressive function of Treg cells, and therefore boosted the anti-cancer immune responses [58]. Thus, these studies suggest opposite roles of YAP and TAZ in Th17/Treg differentiation. It is clear that YAP and TAZ share similar molecular activities [59] and show functional redundancy during embryonic development [3] and regeneration [60] as the simultaneous deletion of YAP and TAZ generally results in a more severe phenotype than their single depletion. However, several lines of evidence in recent years indicate their distinct functions [61,62,63]. This divergence adds an extra level of complexity to the functions of the Hippo pathway in different cell-types in a context-dependent manner. The precise molecular mechanisms underlying these superficially contradictory regulations remain to be elucidated.

### 2.4. Critical Roles of MST1 in B cell Development and Functions

B cell progenitors arise from common lymphoid progenitors (CLP) in the bone marrow, and further develop into mature B cells in the secondary lymphoid tissue, such as the spleen [64] (Figure 3). MST1 is expressed in B-cell lineages and its deficiency causes B cell lymphopenia as well as autoantibody production in humans and in mouse models [29,33,36,39]. MST1-deficient mice show reduced CD19 expression, and disrupted B cell receptor clustering/downstream signaling, which results in a dramatic reduction of B cell viability and a developmental defect of marginal zone B cells (MZB) [65].

As described above, MST1 plays pivotal roles in both T cell and B cell development. Therefore, MST1 deficiency causes T and B cell lymphopenia in human patients, resulting in combined immunodeficiency. However, MST1-deficient patients also exhibit autoimmune-like symptoms, including autoantibody production. These symptoms are likely due to the impaired, MST1-lacking T cell functions, instead of the intrinsic signals in B cells. While T cell-specific MST1-deficient (*Lck*-Cre/*Mst1*^F/F^) mice demonstrated autoantibody production, B cell-specific MST1 deficiency (*Mb1*-Cre/*Mst1*^F/F^) did not result in autoantibody production for up to 18–20 months [66]. As MST1 is critical for the proper activities of FOXO, defective MST1–FOXO signaling impairs the differentiation and function of Treg cells, collapsing immune tolerance and thereby provoking autoimmunity [39,55]. Another study demonstrated that IL-4-rich environments created by MST1-deficient CD4^+^ T cells also contribute towards the uncontrolled B cell responses [67]. Together, these results not only highlight the critical functions of MST1 in B cell development, but also demonstrate its role in maintaining immune tolerance to prevent B cell overactivation through the regulation of CD4^+^ T cells. Despite the critical functions of MST1 in B cell regulation, other components of the Hippo pathway have been poorly explored in B cell lineage. It would be of interest to elucidate the involvement of other Hippo pathway components in B cell development and functions in future studies.

### 2.5. MST1/2 in Dendritic Cell Functions

Dendritic cells arise from bone marrow-resided pre-dendritic cells (pre-DC) that migrate to non-lymphoid (such as payer’s patches and dermis) or lymphoid (mainly spleen) tissues where they further differentiate into lymphoid dendritic cells [68,69] (Figure 3). Differentiated mature dendritic cells in peripheral tissues capture exogenous antigens and migrate to the draining lymph node through CCR7 (C-C motif chemokine receptor 7)-dependent chemotaxis [70]. Among lymphoid dendritic cells, CD8^+^ dendritic cell subset represents the population with a higher capacity of antigen cross-presentation to T cells [71].

An early study found that MST1-deficient mice show impaired trafficking of dendritic cells from the skin to the draining lymph node [42]. MST1-deficient dendritic cells demonstrated reduced attachments to the extracellular matrix in vitro [42]. Another study has suggested that MST1 mediates CCR7-dependent chemotaxis of human mature dendritic cells through the regulation of actin cytoskeleton [72]. Thus, MST1 regulates adhesion and cell motility, orchestrating the efficient migration of dendritic cells. MST1 also plays an important role in directing the T cell lineage by regulating the production of dendritic-cell-derived cytokines. MST1-deficient dendritic cells (*Cd11c*-Cre/*Mst1*^F/F^) produce more IL-6 due to the activation of the p38 MAPK pathway, which in turn stimulates IL-6–STAT3 (signal transducer and activator of transcription 3) signaling in CD4^+^ T cells to facilitate Th17 differentiation [73]. More recently, another study identified MST1/2 as crucial regulators of CD8^+^ dendritic cells that reside in lymphoid tissue and elicit efficient antigen cross-presentation [74]. MST1/2 promote oxidative metabolism and contribute to the maintenance of bioenergetic activities in CD8^+^ dendritic cells. Mechanistically, although not fully elucidated, MST1/2 appear to exert their functions via the alternative Hippo pathway as the dendritic cell-specific ablation of LATS1/2 or YAP/TAZ failed to phenocopy MST1/2 deletion. MST1/2 orchestrate selective expression of the T-cell-activating cytokine IL-12 via crosstalk with the non-canonical NF-κB (nuclear factor-kappa B) signaling pathway in CD8^+^ dendritic cells [74]. Together, these results highlight the important roles of MST1/2 in dendritic cellular functions. Further in vivo analyses involving other Hippo pathway components in each dendritic cell subset will bring advances in our understanding of lineage commitment, differentiation, and the function of dendritic cells.

### 2.6. Hippo Pathway in Macrophages

Macrophages are one of the fastest immune cells that are capable of interacting with antigen at sites of infection or tumor initiation. Macrophage provides both MHC (major histocompatibility complex) class II and peptide complex, together with co-stimulatory signals, to T cells. The production of inflammatory cytokines in response to antigenic stimulation to initiate and maintain the inflammatory response is also a key function of the macrophages [75].

Critical roles of the Hippo pathway in the biological functions of macrophages have been highlighted in recent studies. MST1/2 boost the phagocytic induction of reactive oxygen species and the anti-bacteria response in macrophages. MST1/2 promote the recruitment of mitochondria to phagosome by phosphorylating PKCα (protein kinase C alpha), and thus regulating the mitochondrial trafficking and mitochondrion-phagosome juxtaposition, which is required for effective reactive oxygen species generation to kill bacteria [21]. Intriguingly, several recent studies have revealed the reciprocal interaction between the Hippo pathway and endocytic trafficking [76,77,78,79,80]. It is therefore possible that the Hippo pathway may modulate intracellular trafficking and, in turn, regulate the functions and integrity of organelles in macrophages as well as other immune cell-types.

YAP is also implicated in the regulation of antiviral responses in macrophages. In antiviral immune responses, viral DNA or RNA in cytosol activates TBK1 that phosphorylates and activates the key antiviral transcription factor IRF3 to induce the type I interferon response [81]. YAP binds to IRF3 and thus blocks its dimerization and translocation to the nucleus after viral infection [23]. However, since YAP/TAZ are rarely expressed in a variety of immune cells [24], their regulation in macrophages requires further validation. Another study consistently demonstrated that YAP/TAZ, through their direct binding, impair ubiquitylation-dependent TBK1 activation and antiviral responses in YAP/TAZ-abundant adherent cells, including HEK293, mouse embryonic fibroblasts (MEFs), and NMuMG epithelial cells [24]. In contrast, however, another study suggested that MST1 directly phosphorylates IRF3 to inhibit its dimerization and activation, impeding cytosolic antiviral defense in the aforementioned cell lines [82]. The precise role of the Hippo pathway in regulating antiviral immune response is thus not fully elucidated, but the interplay between canonical (through YAP/TAZ) and the alternative (through IRF3) Hippo pathway downstream of MST1/2 may contribute to these counterintuitive results. Future studies clarifying the role of other Hippo pathway components in antibacterial and antiviral responses will provide a clear mechanistic insight into the functions of the Hippo pathway in innate immune responses.

## 3. Dual Functions of the Intracellular Hippo Signaling in Regulating Immune Responses

As we have highlighted above, the Hippo pathway plays crucial roles in the adaptive immune cell development and functions. Recent studies have revealed that the Hippo pathway in non-immune cells also contributes to the induction and direction of the immune responses in both immunostimulatory and immunosuppressive ways. These involve cytokine production, immune checkpoint molecule expression, and extracellular vesicle release from the non-immune cells, allowing for intercellular communication and orchestrating the immune responses.

### 3.1. Immunostimulatory Role of the Hippo Pathway in Non-Immune Cells

Apart from its physiological roles, recent studies have also suggested the involvement of the Hippo pathway in several aspects of the tumor cell-intrinsic mechanisms of immune suppression in tumor microenvironments (Figure 5). YAP-mediated transcription is shown to promote cytokine/chemokine production in many types of tumors. For example, active YAP in murine prostate adenocarcinoma promotes CXCL5 (C-X-C motif chemokine ligand 5) secretion, which attracts CXCR2 (C-X-C motif chemokine receptor 2)-expressing myeloid derived suppressor cells (MDSC) to suppress the immune responses [83]. Pharmacological inhibition of the CXCL5–CXCR2 axis as well as MDSC depletion by neutralizing antibodies therefore impeded tumor progression [83]. Another study using the KRAS/p53 mutant pancreatic ductal adenocarcinoma model found that active YAP contributes to the differentiation and accumulation of MDSC in tumor microenvironments by promoting the expression and secretion of multiple cytokines and chemokines, including CXCL1/2 and C-C motif chemokine 2 (CCL2) [84]. YAP has further been found to function downstream of the ovarian cancer-specific oncogene PRKCI (protein kinase C iota) to up-regulate TNF (tumor necrosis factor) expression, recruiting MDSC to inhibit cytotoxic T cell functions. [85]. An additional mechanism for YAP-mediated immune-suppression is suggested to be through the recruitment of type II (M2) macrophages that suppress the immune clearance of cancer cells [86]. Tumor-initiating cells with YAP hyperactivation recruit M2 macrophages at the early phases of cancer development, mainly through direct YAP–TEAD targeting of CCL2 and CSF1 (colony stimulating factor 1) [87]. Interestingly, a knockdown of CCL2/CSF1 in tumor-initiating cells blocked M2 macrophage recruitment and abolished tumorigenesis in an immune system-dependent manner, suggesting that YAP-activated tumor-initiating cells are eliminated without the protection of M2 macrophages. Similarly, another in vitro study suggested that YAP in DLD-1 colon cancer cells contributes to the M2 macrophage differentiation of co-cultured THP-1 monocytes [88]. Collectively, these studies demonstrate that the activation of YAP in cancer cells promotes cytokine/chemokine production and thus contributes to the recruitment of immunosuppressive cells in tumor microenvironments.

In addition to the alteration of the cellular composition in tumor microenvironments, a context and cell-type dependent involvement of an immune checkpoint molecule has been implicated in YAP-mediated immune-suppression. The immune checkpoint receptor, programmed cell death 1 (PD-1, also known as PDCD1), and its ligand PD-L1 (programmed death-ligand-1, also known as CD274) provide a negative regulatory pathway that prevents self-antigen recognition by T cells [89]. Cancer cells hijack this built-in regulatory pathway to evade the host immunity by upregulating PD-L1, which results in the apoptosis or anergy of T cells by stimulating suppressive PD-1 signaling in T cells [89]. Several studies suggest that YAP/TAZ suppress T-cell-mediated killing of cancer cells by directly transcribing PD-L1 in human melanoma, lung cancer, and breast cancer cells [90,91,92,93]. YAP–TEAD transcription complex binds to the PD-L1 promoter or enhancer, directly inducing PD-L1 expression. In contrast, another study demonstrated that YAP inhibits IFNγ-inducible PD-L1 expression in murine syngeneic cancer models [94]. YAP–TEAD transcriptional complex induces the expression of its target gene miR-130a, which in turn suppresses the expression of IRF1, the major transcriptional factor for PD-L1 expression, and thus inhibits tumor growth in mice [94]. Immunosuppressive functions of the Hippo pathway will be discussed in the following section.

### 3.2. Immunosuppressive Role of the Hippo Pathway in Non-Immune Cells

It is largely accepted that the Hippo pathway acts as a tumor suppressor in adherent cells that inhibits cell proliferation and survival, preventing tumorigenesis [11,12,13,14]. Inactivation of the Hippo pathway drives a cell-intrinsic program to promote cell proliferation and survival, allowing cellular expansion and migration. Hyperactivation of YAP/TAZ is wide-spread in human neoplasia [8], and numerous reports indicate gene amplification and epigenetic modulation of the YAP/TAZ loci in cancer [12], implying that YAP/TAZ-mediated transcription drives the development and sustainability of human cancer. However, this assumption is in contrast with the fact that germline or somatic mutations in key components of the canonical Hippo pathway are relatively rare in human cancers [11,12,13,14]. So far mutations in NF2 or LATS2 loci have been highlighted in specific types of human cancers, such as mesothelioma, schwannoma, and meningioma. However, another conundrum is that these mutations only occur in specific cancer histotypes and are not broadly distributed [13]. Therefore, it is not surprising that the Hippo pathway has divergent functions in cancer growth. For instance, YAP has been shown to have a tumor suppressive function in colorectal cancer [95], multiple myeloma [96], breast cancer [97,98], and lung squamous cell carcinoma [99]. Moreover, the cell type-dependent functions of LATS1/2 in promoting or suppressing cancer cell growth became apparent [100]. These studies suggest that the precise role of the Hippo pathway in human cancer is context dependent, and that the Hippo pathway has dual functions in both cancer progression and suppression.

Another, perhaps more feasible, possibility for the low mutation rate of the Hippo pathway components in human neoplasia is that this pathway may possess built-in feedback mechanisms that prevent the overgrowth of undesirable cells in the organism. These include cell-intrinsic feedback mechanisms that prevent the Hippo pathway dysregulation [101,102,103], the selective elimination of YAP-activated cells from the neighboring cells [104], and immunomodulation by the Hippo pathway (Figure 5) [94,105]. As mentioned above, YAP inhibits IFNγ-inducible PD-L1 expression and thus inhibits tumor growth in syngeneic mouse models [94]. Another study revealed that LATS1/2-deficient cancer cells induce a type I interferon response via the host TLR signaling, enhancing cross-presentation of tumor antigens to boost the anti-cancer immune response [105]. While LATS1/2 deficiency showed a significant increase in anchorage-independent cancer cell growth in vitro, their growth in vivo in immune-competent mice is severely compromised due to the induction of strong anti-cancer immune responses. Though the precise mechanism remains to be elucidated, it is likely that nucleic-acid-rich extracellular vesicles (EVs), released from LATS1/2-deficient cancer cells, stimulate the host nucleic-acid-sensing TLR signaling to induce a type I interferon response [105]. Therefore, enhanced immunogenicity unmasked by the LATS1/2 deletion in cancer cells induces strong immune responses and overwhelms any growth advantage that might be gained due to LATS1/2 deletion, leading to a strong inhibition of tumor growth in the immune-competent host. Thus, collectively, immunosuppressive functions of the Hippo pathway provide a built-in homeostatic control mechanism that prevents tissue overgrowth and tumorigenesis. Interestingly, while genetic ablation of MST1/2 in liver [106,107,108] and intestine [109] promotes tumorigenesis, deletion of LATS1/2 in liver [110,111] or kidney [112] does not result in the development of cancer. These observations imply that the molecular functions of MST1/2 and LATS1/2 are not necessarily the same, and canonical and alternative Hippo pathways in non-immune cells may exert different functions on immunomodulation in a context-dependent manner. This complexity adds to the already divergent functions of the Hippo pathway in the regulation of the mammalian immune system.

## 4. Conclusions

In recent years, much attention has been drawn toward the elucidation of the diverse roles of Hippo signaling in adaptive immunity. Especially, MST1/2 have been revealed to be broadly involved in maintaining proper adaptive immune responses by regulating cell survival, differentiation, migration, and function in diverse immune cell types. It is interesting to note that other key Hippo pathway components, such as LATS1/2 and YAP/TAZ, seem to be less important and only partially participate in these processes, as is evident by their negative expression or unaffected phosphorylation status upon immune cell activation [29,57]. Moreover, studies utilizing conditional deletion of YAP/TAZ or LATS1/2 in mice revealed that these genes are dispensable for some immune cell functions [28,74]. Although it requires further validation, previous studies thus imply that the alternative Hippo signal transduction may play key roles along with MST1/2 in regulating immune cell functions in several contexts. This includes the characterization of NDR1/2 in T cell biological functions [44,113]. NDR1/2 (also known as STK38/STK38L) belong to the NDR/LATS subfamily of AGC serine/threonine kinases. NDR1/2 and LATS1/2 share many overlapping molecular functions and regulatory mechanisms, such as phosphorylation by MST1/2 kinases, binding to MOB1A/B, and kinase activity towards YAP (reviewed in [114]). In addition, unbiased proteomic studies of the Hippo pathway interactome have consistently placed NDR1/2 in the Hippo pathway network [76,77,79]. Therefore, in future studies, it is worth validating whether NDR1/2 act as the main mediators of the alternative Hippo pathway in vivo and investigating how these diverse regulations of the Hippo pathway result in different biological outputs downstream of NDR1/2- or LATS1/2-dependent signaling branches in the adaptive immune system.

Accumulating evidence, using semisynthetic substrates and model systems, suggests that the Hippo pathway involves important mechano-regulated factors that integrate physical cues to gene expression and cellular responses. Mechanical signals such as cell shape, extracellular matrix stiffness, and shear flow, modulate activities of RHO GTPases, which in turn lead to actin cytoskeleton remodeling (reviewed in [115]). The actin cytoskeleton network can control nuclear–cytoplasmic shuttling and the transcriptional activities of YAP/TAZ [116,117,118]. In particular, focal adhesion regulates the Hippo signaling through RAP2 GTPases [119]. The integrin signaling also transduces mechano-signals to the Hippo pathway [120,121,122,123,124]. Given that the immune cells are continuously exposed to stresses that came from extracellular matrix and liquid shear-flow, it is of particular interest to investigate the involvement of the Hippo pathway in sensing mechanical forces and processing physical cues in lymphocytes. While we have made substantial advances in our understanding of the Hippo signaling in the adaptive immune system, the research must continue in order to broaden our knowledge on the interplay between the Hippo signaling pathway and the immune system. Delineating these interactions will have important clinical implications in autoimmune diseases and cancer.

## Figures and Tables

**Figure 1 cells-08-00398-f001:**
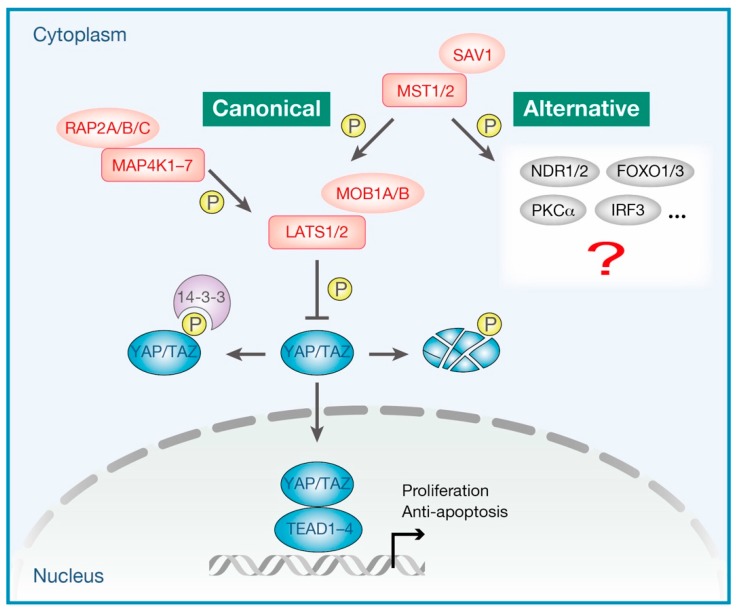
Canonical and alternative Hippo pathways. The heart of the Hippo pathway consists of the kinase module (indicated in red) and the transcriptional module (indicated in blue). The kinase module includes 11 kinases, namely mammalian STE20-like protein kinases (MST1/2), mitogen-activated protein kinase kinase kinase kinases (MAP4K1-7), and large tumor suppressor kinases (LATS1/2), as well as their activating adaptor proteins, salvador family WW domain-containing protein 1 (SAV1), Ras-related proteins (RAP2A/B/C), and MOB kinase activators (MOB1A/B). The transcriptional module includes the transcriptional co-activators, namely, yes-associated protein (YAP) and transcriptional co-activator with PDZ-binding motif (TAZ), and in addition, the transcription factors TEA domain family members (TEAD1–4). When the upstream signals are integrated to activate the Hippo pathway, LATS1/2 kinases phosphorylate and inhibit YAP/TAZ. Phosphorylation of YAP/TAZ promotes their proteasomal degradation or cytoplasmic retention via 14-3-3 binding. In contrast, when the kinase module is inactivated (the Hippo pathway is inactivated), hypophosphorylated YAP/TAZ translocate into the nucleus wherein they bind to TEAD1–4 and thus induce proliferative and anti-apoptotic gene transcription. In this review, we define the signaling that specifically regulates LATS1/2 kinase activity and/or YAP/TAZ transcriptional activity as the “canonical” Hippo pathway. The other signaling cascades that involve MST1/2 but do not regulate LATS1/2 kinases or YAP/TAZ are defined as the “alternative” Hippo pathway. MST1/2 have been shown to modulate a number of proteins, including nuclear Dbf2-related kinases (NDR1/2, also known as STK38/STK38L), forkhead box O (FOXO1/3), protein kinase C alpha (PKCα), and interferon regulatory factor 3 (IRF3) in the regulation of immune system.

**Figure 2 cells-08-00398-f002:**
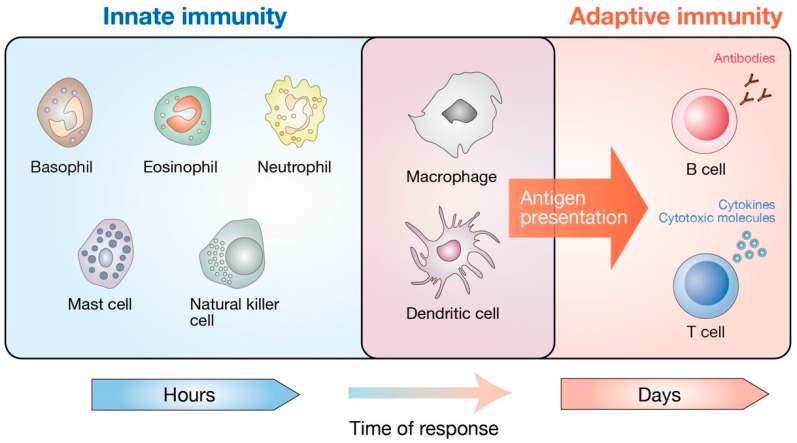
Cellular components of the mammalian immune system. The mammalian immune system consists of two distinct parts, innate and adaptive immunity. Basophils, eosinophils, neutrophils, mast cells, natural killer cells, macrophages, and dendritic cells mediate the innate immunity. They provide the first line of defense against bacteria, viruses, and cancer. The adaptive immune system refers to an antigen-specific defense mechanism that takes several days to develop but provides long-lasting protection. The adaptive immune system includes B cell-mediated humoral immunity and T cell-mediated cellular immunity, both of which are directed towards the specific antigens. Macrophages and dendritic cells are unique subsets that have both innate and adaptive immune cell traits. As professional antigen-presenting cells, macrophages and dendritic cells are critical in the induction of adaptive immunity by presenting the antigens to antigen-specific T and B lymphocytes.

**Figure 3 cells-08-00398-f003:**
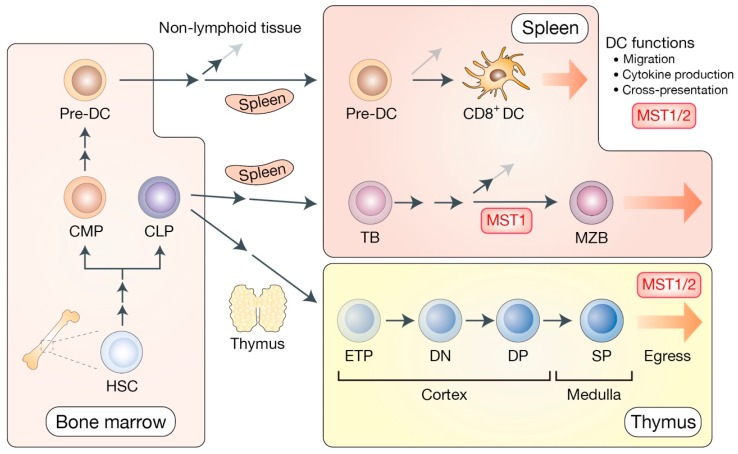
The Hippo pathway in the development of dendritic cells, B cells, and T cells. In bone marrow, hematopoietic stem cells (HSC) give rise to common myeloid progenitors (CMP) and common lymphoid progenitors (CLP). CMP generate pre-dendritic cells (pre-DC). Pre-DC are dendritic cell-restricted progenitors that routinely leave bone marrow for the lymphoid and non-lymphoid tissues to get differentiated into local dendritic cells. In the spleen, pre-DC further maturate into CD8^+^ DC that possess a higher capacity of antigen cross-presentation to T cells. CLP give rise to transitional B cells (TB) that differentiate into marginal zone B cells (MZB) in the spleen. MZB provide the first line of defense against specific pathogens by mediating rapid antibody responses. CLP in the bone marrow migrate into the thymic cortex to generate early thymic progenitors (ETP), also known as double negative 1 (DN1) thymocytes. After the DN2–4 maturation stage, they subsequently become double positive (DP) thymocytes. DP thymocytes further mature into single positive (SP) T cells that migrate into the thymic medulla and egress to the periphery. MST1/2 are shown to mediate DC functions, B cell development, and the thymic egress of mature T cells via distinct mechanisms.

**Figure 4 cells-08-00398-f004:**
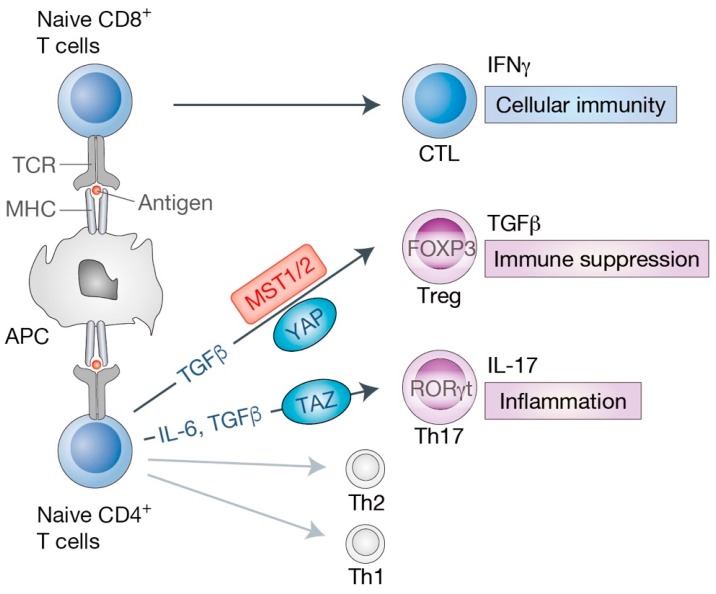
Roles of the Hippo pathway in effector T cell differentiation. The primary event of naive T cell activation is the interaction between T cell receptor (TCR) on T cells and an antigen loaded on antigen-presenting cells (APC). APC present antigenic peptides with major histocompatibility complex (MHC) class I (for CD8^+^ T cells) or class II (for CD4^+^ T cells) molecules. Upon activation, naive CD8^+^ T cells differentiate into cytotoxic T lymphocytes (CTL) that mediate cellular immunity and secrete series of cytokines such as interferon gamma (IFNγ). Naive CD4^+^ T cells differentiate into T helper (Th) cells, including effector T cells (Th1, Th2, Th17) and regulatory T cells (Treg). Transforming growth factor beta (TGFβ) signaling is required for Treg lineage, while interleukin 6 (IL-6) together with TGFβ cytokines are essential for Th17 differentiation. Key transcription factors have also been identified for each T cell lineage [forkhead box P3 (FOXP3) for Treg, RAR related orphan receptor gamma t (RORγt, also known as RORC) for Th17]. Treg cells produce TGFβ and suppress effector T cell activities, maintaining immune system homeostasis and self-tolerance. Th17 cells produce IL-17 cytokine and are involved in chronic and autoimmune inflammation. MST1/2 and YAP are shown to mediate Treg differentiation by diverse mechanisms. TAZ acts as a critical co-activator of RORγt for Th17 differentiation.

**Figure 5 cells-08-00398-f005:**
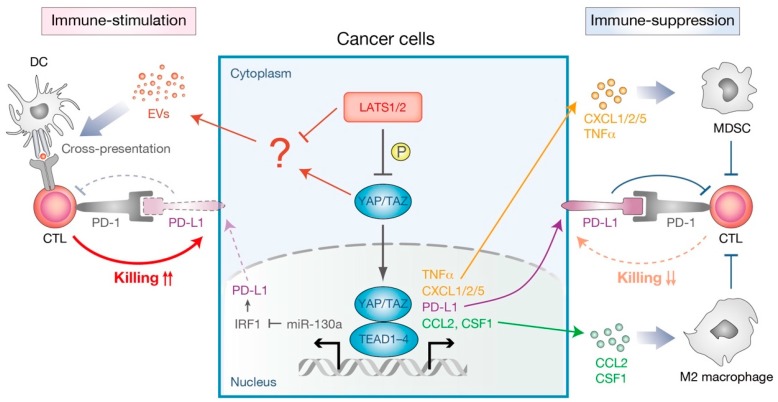
Dual functions of the cancer cell-intrinsic Hippo pathway in immune modulation. (Left) Immunnostimulatory effects mediated by YAP activation or LATS1/2 inactivation. In cancer cells, YAP inhibits IFNγ-inducible PD-L1 (programmed death-ligand-1, also known as CD274) expression partially through miR-130a-mediated suppression of IRF1. IRF1 up-regulates the expression of PD-L1 that binds to PD-1 (programmed cell death 1, also known as PDCD1) on T cells and provides inhibitory signals to cytotoxic T cells (CTL). Upon deletion of LATS1/2, extracellular vesicles (EVs) secreted from cancer cells trigger an anti-cancer immune response by stimulating the host nucleic-acid-sensing pathway and enhancing antigen cross-presentation. (Right) Immunosuppressive effects mediated by YAP activation. Active YAP induces the expression of PD-L1, as well as cytokines [TNFα (tumor necrosis factor alpha), CSF1 (colony stimulating factor 1)] and chemokine ligands [CXCL1 (C-X-C motif chemokine ligand 1), CXCL2, CXCL5, and CCL2 (C-C motif chemokine 2)]. Those cytokines or chemokines recruit immunosuppressive cells, such as myeloid derived suppressor cells (MDSC) and M2-type macrophages, to inhibit CTL functions.

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
