# Peer review of "Hippo Pathway in Mammalian Adaptive Immune System"

_cells, 2019, doi:10.3390/cells8050398_

Round 1
Reviewer 1 Report
The manuscript is well written and describes the role of Hippo signaling in the mammalian adaptive immune system- a timely compilation of evidences on this relatively new aspect of Hippo function.
The illustrations are well organized and figures/cartoons adequately describe the various aspects of Hippo interactions with the immune system and provides a much needed summary of developments in this field.
Author Response
Response to Reviewer #1
The manuscript is well written and describes the role of Hippo signaling in the mammalian adaptive immune system- a timely compilation of evidences on this relatively new aspect of Hippo function. The illustrations are well organized and figures/cartoons adequately describe the various aspects of Hippo interactions with the immune system and provides a much needed summary of developments in this field.
We thank the reviewer for careful review and the positivecomments on our manuscript.

Reviewer 2 Report
The manuscript entitled "Hippo pathway in mammalian adaptive immune system" by Yamauchi and Moroishi summarized current understanding of the emerging roles of Hippo pathway in adaptive immune cell development and differentiation. The authors also highlighted the recent findings of the dual functions of Hippo pathway in autoimmunity and anti-cancer immune responses.
Overall, the manuscript is well written. There are some minor points that could be addressed to further improve the presentation of this work.
1. The figure legend title of figure 1 and 3 need to be changed. For example: figure 1 mainly described the regulation of Hippo pathway, but not as authors described “Regulation of adaptive immune cells by canonical and alterative Hippo pathway”; figure 3 mainly described the development of dendritic cells, B cells and T cells.
2. Line 436, the reference was not formatted.
Author Response
Response to Reviewer #2
The manuscript entitled "Hippo pathway in mammalian adaptive immune system" by Yamauchi and Moroishi summarized current understanding of the emerging roles of Hippo pathway in adaptive immune cell development and differentiation. The authors also highlighted the recent findings of the dual functions of Hippo pathway in autoimmunity and anti-cancer immune responses. Overall, the manuscript is well written. There are some minor points that could be addressed to further improve the presentation of this work.
We thank the reviewer for the positive comments on our manuscript and helpful suggestions. Our specific responses to the points raised are as follows:
1) The figure legend title of figure 1 and 3 need to be changed. For example: figure 1 mainly described the regulation of Hippo pathway, but not as authors described “Regulation of adaptive immune cells by canonical and alterative Hippo pathway”; figure 3 mainly described the development of dendritic cells, B cells and T cells.
(Response) We thank the reviewer for pointing out those mistakes. For Figure 1, we have changed the title to “Canonical and alternative Hippo pathway” (line 119). For Figure 3, instead of changing the title, we have included MST1/2 in the figure to explain the role of the Hippo pathway in dendritic cell functions, B cell development, and thymic egress of mature T cells (line 214, new Figure 3).
2) Line 436, the reference was not formatted.
(Response) We appreciate the careful review of the manuscript. Because the corresponding reference has been posted on bioRxiv but not yet been peer-reviewed, we decided to remove the reference and sentence from the revised manuscript (line 642).

Reviewer 3 Report
Hippo pathway in mammalian adaptive immune system.
Yamauchi and Moroishi.
The authors review the role of the core proteins of the Hippo pathway in immune adaptive response. In general, the review is informative and cover many of the aspects known about this area. the authors include relevant and up to date references to support the information included. In particular they give extensive evidence showing that MST1 and MST2 seem to have an important role in this biological function while the other proteins LATS1, YAP and TAZ seem less important. The review is appropriated for the journal readership with some reviews and editing (see below for some examples). However, there are some points that could contribute to make the review more useful for a wider audience not familiar with hippo signalling. Also, some statements can be argued to be not true or in need of further explanation.
1. Thus, although the authors mention it extensively, it is surprising that in figure 1 they do not actually include what they call the alternative hippo signalling, while they show the so-called canonical pathway. It would be more informative for the reader not familiar with the Hippo signalling network to include the modules that are not part of the canonical module but are discussed in the text. Even more, I would strongly suggest inclusion of well documented crosstalk between this pathway and other signalling networks (AKT, MAPK, p53, FOXO) which could help the reader to integrate the information given with other knowledge about adaptive immune system (similar to what they somehow do in figure 4 and 5).
2. The authors make a point about the relative low number of mutations of Hippo signalling proteins. This is only true if they focus on the canonical pathway, which as shown in this review is a very limited approach to the functions of these proteins. If they were to include proteins such as RASSF1, p53, FOXO, ERBB, AKT and RAF this statement is not sustained and makes this network highly deregulated in cancer (doi: 10.3390/genes7060028).
3. Line 242 the statement “Although YAP and TAZ predominantly show functional redundancy” is not supported by the literature. These proteins share a certain degree of redundancy but there are growing evidence were differences have been described (ie doi: 10.1074/jbc.RA118.002715, doi: 10.1002/stem.2652, doi: 10.1002/path.4745.)
4 .Line 410: “For instance, YAP has been shown to have a tumor suppressive function in colorectal cancer [85], multiple myeloma [86], and lung squamous cell carcinoma [87].” It is correct (and important), to see this included here to move from the narrow view imposed by the canonical view of the Hippo signalling, but the authors must include reference to the works in breast cancer were this tumour suppressor role is well supported and indicate that the tumour suppressor role of YAP is related to non-canonical Hippo signalling (i.e. 10.1016/j.molcel.2007.08.008, 10.1038/cdd.2008.108, 10.1038/cdd.2008.108)
Minor points:
5. “Figure 3. The Hippo pathway in the development of dendritic cells, B cells, and T cells.” The title and the content of the figure do not match, since there is not explanation or the function of the pathway in the figure or the legend. The authors should change the tittle accordingly although it would be very informative to have some of the changes of the proteins of the hippo pathway that are described in the text included in the figure (similar to what they do in figure 4)
Some examples of sentences in need of review (but not limited to these)
6. Line 150 suggesting a key functions. Function
7. Line 261 On the one hand, T cell-specific MST1-deficient (Lck-Cre/Mst1F/F) mice demonstrated autoantibody production, and on the other hand, B cell-specific MST1 deficiency (Mb1-Cre/Mst1F/F) did not show autoantibody production for up to 18–20 months.
This sentence is confusing and has too many hands.
8. Line 436: this sentence needs editing to include the reference and missing article “does not develop tumor. Similarly, loss of LATS1/2 in adult pancreatic acinar cells induces pancreatic inflammation [REF:https://doi.org/10.1101/522938]. These observations imply that the molecular functions of MST1/2 and LATS1/2 are not necessarily [the] same,”
Author Response
Response to Reviewer #3
The authors review the role of the core proteins of the Hippo pathway in immune adaptive response. In general, the review is informative and cover many of the aspects known about this area. the authors include relevant and up to date references to support the information included. In particular they give extensive evidence showing that MST1 and MST2 seem to have an important role in this biological function while the other proteins LATS1, YAP and TAZ seem less important. The review is appropriated for the journal readership with some reviews and editing (see below for some examples). However, there are some points that could contribute to make the review more useful for a wider audience not familiar with hippo signalling. Also, some statements can be argued to be not true or in need of further explanation.
We thank the reviewer for the positive comments on our manuscript and many constructive suggestions, which we feel have helped us to greatly improve the manuscript. Our specific responses to the points raised are as follows:
1) Thus, although the authors mention it extensively, it is surprising that in figure 1 they do not actually include what they call the alternative hippo signalling, while they show the so-called canonical pathway. It would be more informative for the reader not familiar with the Hippo signalling network to include the modules that are not part of the canonical module but are discussed in the text. Even more, I would strongly suggest inclusion of well documented crosstalk between this pathway and other signalling networks (AKT, MAPK, p53, FOXO) which could help the reader to integrate the information given with other knowledge about adaptive immune system (similar to what they somehow do in figure 4 and 5).
(Response) We agree with the reviewer and have now included the alternative components discussed primarily in the text (line 118, new Figure 1). According to the reviewer’s suggestion, we also described the crosstalk between the Hippo pathway and other pivotal signaling networks involved in immune regulation (lines 88–90).
2) The authors make a point about the relative low number of mutations of Hippo signalling proteins. This is only true if they focus on the canonical pathway, which as shown in this review is a very limited approach to the functions of these proteins. If they were to include proteins such as RASSF1, p53, FOXO, ERBB, AKT and RAF this statement is not sustained and makes this network highly deregulated in cancer (doi: 10.3390/genes7060028).
(Response) We agree with the reviewer and have now added the word “canonical” to the corresponding sentence to avoid misleading statement (line 610). As mentioned above, we also discussed the signaling network surrounding the Hippo pathway and cited the suggested reference (doi: 10.3390/genes7060028) (lines 88-90).
3) Line 242 the statement “Although YAP and TAZ predominantly show functional redundancy” is not supported by the literature. These proteins share a certain degree of redundancy but there are growing evidence were differences have been described (ie doi: 10.1074/jbc.RA118.002715, doi: 10.1002/stem.2652, doi: 10.1002/path.4745.)
(Response) We appreciate the reviewer pointed this out. We have revised the corresponding sentences to more accurately describe both overlapping and distinct functions of YAP and TAZ in the revised manuscript (lines 366-370).
4) Line 410: “For instance, YAP has been shown to have a tumor suppressive function in colorectal cancer [85], multiple myeloma [86], and lung squamous cell carcinoma [87].” It is correct (and important), to see this included here to move from the narrow view imposed by the canonical view of the Hippo signalling, but the authors must include reference to the works in breast cancer were this tumour suppressor role is well supported and indicate that the tumour suppressor role of YAP is related to non-canonical Hippo signalling (i.e. 10.1016/j.molcel.2007.08.008, 10.1038/cdd.2008.108, 10.1038/cdd.2008.108)
(Response) We apologize for the omission of those papers. We have now cited the relevant papers and highlighted this in the revised manuscript (line 616).
Minor points:
5) “Figure 3. The Hippo pathway in the development of dendritic cells, B cells, and T cells.” The title and the content of the figure do not match, since there is not explanation or the function of the pathway in the figure or the legend. The authors should change the tittle accordingly although it would be very informative to have some of the changes of the proteins of the hippo pathway that are described in the text included in the figure (similar to what they do in figure 4)
(Response) According to the reviewer’s suggestion, we have included MST1/2 in the figure to explain the role of the Hippo pathway in dendritic cell functions, B cell development, and thymic egress of mature T cells (line 214, new Figure 3).
Some examples of sentences in need of review (but not limited to these)
We apologize for typos and grammar mistakes. The manuscript has now been checked by a professional English editing service.
6) Line 150 suggesting a key functions. Function
(Response) Corrected (line 210).
7) Line 261 On the one hand, T cell-specific MST1-deficient (Lck-Cre/Mst1F/F) mice demonstrated autoantibody production, and on the other hand, B cell-specific MST1 deficiency (Mb1-Cre/Mst1F/F) did not show autoantibody production for up to 18–20 months. This sentence is confusing and has too many hands.
(Response) We have now carefully revised the manuscript (lines 386-388).
8) Line 436: this sentence needs editing to include the reference and missing article “does not develop tumor. Similarly, loss of LATS1/2 in adult pancreatic acinar cells induces pancreatic inflammation [REF:https://doi.org/10.1101/522938]. These observations imply that the molecular functions of MST1/2 and LATS1/2 are not necessarily [the] same,”
(Response) We appreciate the careful review of the manuscript. Because the corresponding reference has been posted on bioRxiv but not yet been peer-reviewed, we decided to remove the reference and sentence from the revised manuscript (line 642).